# Electric activity at magnetic moment fragmentation in spin ice

D. I. Khomskii [1✉]

Spin ice systems display a variety of very nontrivial properties, the most striking being the existence in them of magnetic monopoles. Such monopole states can also have nontrivial electric properties: there exist electric dipoles attached to each monopole. A novel situation is encountered in the moment fragmentation (MF) state, in which monopoles and antimonopoles are perfectly ordered, whereas spins themselves remain disordered. We show that such partial ordering strongly modifies the electric activity of such systems: the electric dipoles, which are usually random and dynamic, become paired in the MF state in ($\mathbf{d}$, $-\mathbf{d}$) pairs, thus strongly reducing their electric activity. The electric currents existing in systems with non-coplanar spins are also strongly influenced by MF. We also consider modifications in dipole and current patterns in magnetic textures (domain walls, local defects) and at excitations with nontrivial dynamics in a MF state, which show very rich behaviour and which could in principle allow to control them by electric field.

[1] II. Physikalisches Institut, Universität zu Köln, Köln, Germany. ✉email: khomskii@ph2.uni-koeln.de

Close connection between electric and magnetic phenomena is a cornerstone of modern physics, going back to the work of James Clerk Maxwell, and, even earlier on, to that of Michael Faraday. Recently it acquired novel significance in the big field of spintronics, which involves such phenomena as multiferroics, topological systems, Rashba spin–orbit coupling, etc.[1–3]. The magnetic properties of systems with localised electrons are very diverse. There are different types of magnetic ordering—ferromagnetism, antiferromagnetism, spiral structures, and, similarly, different types of spin liquids, with their eventual topological features. A very interesting phenomenon was discovered in spin ice systems such as some kagome and pyrochlore materials[4]—the appearance of states resembling the famous magnetic monopoles[5]. This discovery led to a flurry of activity, both theoretical and experimental[6–8].

It was later realised that for magnetic monopoles in spin ice there is an interesting interplay between magnetic and electric degrees of freedom: it was shown that on each magnetic monopole there exists an electric dipole attached to it[9], see Fig. 1. This prediction was later followed up by very interesting theoretical development[10–12], and confirmed by different experimental means[13,14]. Thus the interplay between electricity and magnetism was also demonstrated for magnetic monopoles.

Recently, an interesting new twist in the monopole story appeared: it was shown that there occurs in these systems a novel state in which magnetic monopoles exist not only as excitations, but can be also present in the ground state. Especially interesting is the state in which magnetic monopoles and antimonopoles exist on each metal tetrahedra in pyrochlores and are ordered in a two-sublattice fashion (the tetrahedra in pyrochlores and at B-sites of spinels form a bipartite diamond lattice). At the same time the spins themselves remain disordered, see Fig. 2 below.

This novel state with such partial ordering was shown to be a consequence of magnetic moment fragmentation (MF)[15,16] (or spin fragmentation state—not to be confused with spin fractionalization): by the Helmholtz decomposition the magnetization can be divided into divergence-free and divergence-full components, and the monopoles and antimonopoles, i.e. magnetic charges, the sources of the divergent-full part of magnetisation, are ordered. As is clear from ref. [15], such decomposition of total magnetization into divergence-free (Coulomb phase) and divergence-full part is valid even for random monopoles. However, in the follow-up literature the terminology 'magnetic fragmentation' is always used for the state with ordered monopoles and antimonopoles at every site (every tetrahedron in pyrochlores or every triangle in kagome systems). In this paper we will also use the term MF in this sense.

This partially-ordered state has a very clear manifestation in magnetic neutron scattering: the Bragg scattering due to ordered monopoles (actually forming the (all-in)–(all-out) state) coexists with strong diffuse scattering with pinch-points due to disordered spins[17,18]. This state can exist in pyrochlores for a certain range of parameters and temperatures[19]; it can be stabilised by the staggered field present in specific situations[20,21].

A similar state with the ordering of monopoles–antimonopoles exists also in kagome systems, Fig. 2b; actually this state was predicted for kagome systems[22,23] even before the idea of moment fragmentation in pyrochlores was proposed. In kagome spin ice systems with moment fragmentation the monopoles and antimonopoles are ordered into two sublattices in a bipartite

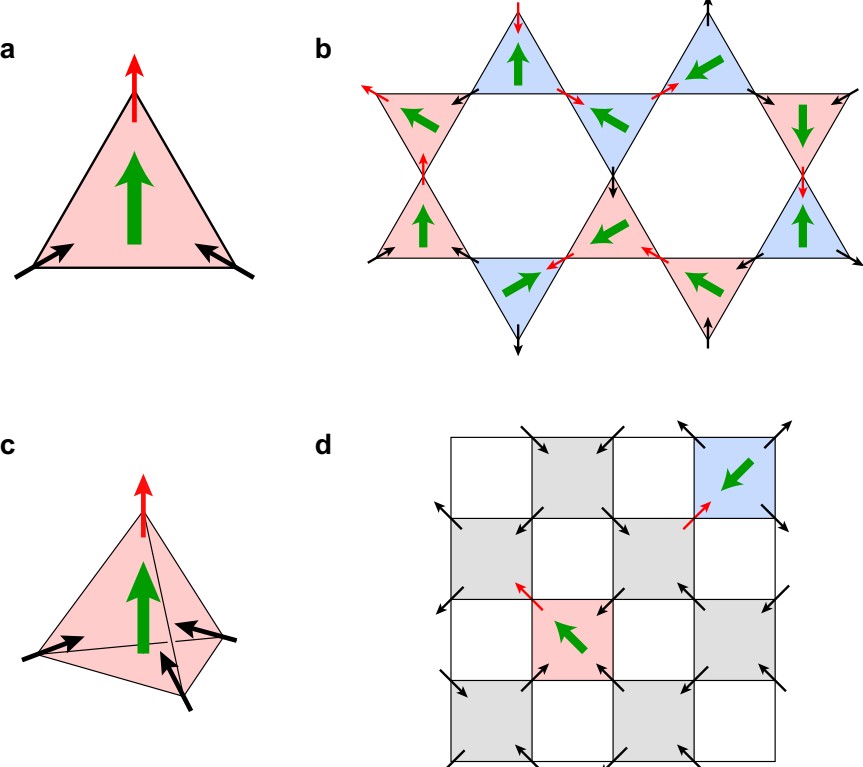

**Fig. 1 The appearance of an electric dipole on a magnetic monopole. a** The triangle (a building block of kagome ice). The dipole points towards the 'special' spin (red arrow). **b** Spin ice kagome lattice with random monopoles ($\mu$) and antimonopoles ($\bar{\mu}$) and random dipoles. **c** The tetrahedron (the building block of pyrochlores) with a monopole. The dipole points to the 'special' spin (red arrow): the out-spin in the monopole with the (3-in)–(1-out) configuration, and the in-spin in the antimonopole with the (1-in)–(3-out) configuration. **d** A typical configuration of pyrochlore spin ice with random monopoles ($\mu$) and antimonopoles ($\bar{\mu}$) and the electric dipoles associated with them shown on a schematic representation of this state on a 2d plot. The monopoles are marked by pink and antimonopoles by blue colour; electric dipoles are shown by thick green arrows.

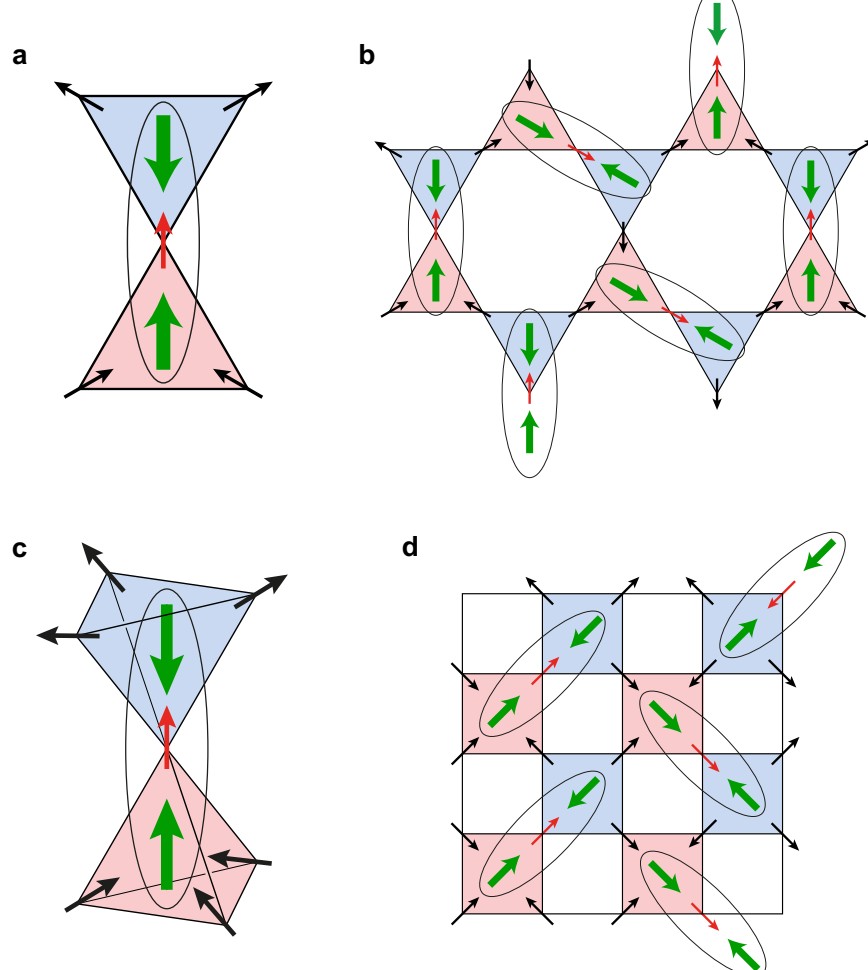

**Fig. 2 Moment fragmentation state and electric dipoles in this state in kagome and pyrochlore spin ice. a** Monopole--antimonopole pair (the main building blocks of kagome system with moment fragmentation) with common 'special' spin (red arrow), demonstrating the pairing of opposite dipoles (formation of (**d**, −**d**) pairs). **b** Example of a moment fragmentation state in kagome spin ice with ordered monopoles, disordered spins and paired dipoles. **c** Pair of tetrahedra with monopole--antimonopole with a common 'special' spin, demonstrating the formation of paired (**d**, −**d**) dipoles. **d** An example of moment fragmentation state in a pyrochlore system with ordered monopoles, disordered spins and paired dipoles. The (**d**, −**d**) pairs are marked by ovals. Colour coding is the same as in Fig. 1.

honeycomb lattice of triangles. The kagome ice state can be also realized in pyrochlores such as $Dy_2Ti_2O_7$ and $Ho_2Ti_2O_7$ in a [111] magnetic field slightly below the transition to the fully-ordered state[24], and it is rarely noticed that this state has moment fragmentation with monopole ordering in kagome layers, see Supplementary Information 1.

Spin-ice states in kagome systems are intrinsically the states with monopoles and antimonopoles in the ground state: every triangle has either (2-in)—(1-out) spins (monopole, $\mu$, with magnetic charge $Q = +1$), or (1-in)—(2-out) (antimonopole, $\bar{\mu}$, $Q = -1$). That is why the conditions of getting moment fragmentation in kagome systems is easier to achieve than in pyrochlores, in which one first has to create monopoles and antimonopoles, and after that to order them. Nevertheless the resulting properties of these systems are in many respects similar, and in what follows we will first consider our effects—the electric properties of moment fragmentation state—on the example of kagome spin-ice systems, and turn to pyrochlores later.

The question we want to address is what happens with charge degrees of freedom of monopoles, such as electric dipoles, and also with electric currents and orbital moments, when the system goes to the moment fragmentation state with ordered monopoles

and antimonopoles but with disordered spins. The second question we want to consider is the electric activity of excitations and defects, e.g. domain walls (DW) in a moment fragmentation state, which turns out to be quite nontrivial.

## Results

We will first quickly recapitulate the arguments leading to the appearance of electric dipoles **d** attached to magnetic monopoles. As was shown in[25,26], proceeding from the Hubbard model, in a magnetic texture of triangles with spins $S_1$, $S_2$ and $S_3$ there appears spontaneous charge redistribution (even in a simple strong Mott insulator with one electron per site and with $U/t \gg 1$), so that the charges at different sites are not equal to $e = 1$, but, for example, the charge at site 1 is

$$e_1 \sim 1 + b[S_1 \cdot (S_2 + S_3) - 2S_2 \cdot S_3], \qquad (1)$$

where the coefficient $b$ in the Hubbard model is given by $b = 8t^3/U^2$. Consequently, if this spin correlation function is nonzero, there would appear in such a triangle an electric dipole moment.

One can easily see that the monopoles and antimonopoles in triangles of kagome ice always have such dipoles at every triangle, with the dipole moment pointing in the direction of the 'special'

spin—the out-spin in the (2-in)–(1-out) monopole, and the in-spin in the antimonopole with the (1-in)–(2-out) spin configuration. For pyrochlore systems the (2-in)–(2-out) states of a tetrahedron (as well as the (all-in)–(all-out) states) do not have such dipoles, whereas they are present in monopole and antimonopole configurations, Fig. 2. The dipole would in this case also point in the direction of the 'special' spin (out-spin in the (3-in)–(1-out) monopole configuration, in-spin in the antimonopole).

In addition to electric dipoles, always accompanying magnetic monopoles[9], we will also consider electric currents, which are present in systems with magnetic triangles when the magnetic structure is noncoplanar. As was shown in[25,26], in this case there exist in such a triangle with spins $\mathbf{S}_1$, $\mathbf{S}_2$ and $\mathbf{S}_3$ a real circular electric current

$$j_{123} = c\kappa(123) \tag{2}$$

where $\kappa$ is the scalar spin chirality,

$$\kappa(123) = \mathbf{S}_1 \cdot [\mathbf{S}_2 \times \mathbf{S}_3] \tag{3}$$

and the coefficient $c$ in the nondegenerate Hubbard model is given by $c = 24et^3/\hbar U^2$ [25]. In most spin ice systems the spins are intrinsically noncoplanar, thus one can expect the appearance of such currents in triangles in some kagome systems and in tetrahedra in pyrochlores. The question is thus what would become of these currents (and the corresponding orbital moments $\mathbf{L} \sim \mathbf{j}$), and of course with electric dipoles, when the system goes into the moment fragmentation state. As the effects connected with these currents and with the respective orbital moments have less apparent experimental consequences, we will present in the main text only the corresponding results, the detailed treatment being deferred to Supplementary Information. Note that one often discusses the fictitious magnetic moment or magnetic flux, accompanying the states with nonzero scalar spin chirality due to Berry phase mechanism, see e.g.[27]. But it is interesting and not usually appreciated, that there appear in this case also real currents and real orbital magnetic moments proportional to scalar spin chirality in[25].

**Kagome systems**. In kagome spin ice, with monopoles and antimonopoles at every triangle (disordered and mobile in the usual case) the electric dipoles exist already in the ground state. But in the usual situation, without MF, they are random and fluctuating, Fig. 1b.

This, however, strongly changes if we go to the moment fragmentation state with ordered monopoles and antimonopoles. In this case each triangle with the monopole $\mu$ is surrounded by three triangles with antimonopoles $\bar{\mu}$. One immediately sees that in this case there would always exist monopole–antimonopole ($\mu$–$\bar{\mu}$) pairs with the common spin being this 'special' spin for both triangles, Fig. 2a. But as explained above, in this case the electric dipoles in these triangles would always point towards each other and would be 'paired' into a state ($\mathbf{d}$, $-\mathbf{d}$) without net dipole moment (but with nonzero quadrupole moment), see Fig. 2a.

If, however, we consider the $\mu$–$\bar{\mu}$ pair with different 'special' spins, the dipole moments on these triangles would point in different directions. But every such triangle would then have another 'mate', another triangle having this common special spin, having an opposite dipole, so that in effect in this case all dipoles would be paired in ($\mathbf{d}$, $-\mathbf{d}$) pairs, Fig. 2b. This is actually the main effect we want to stress: the monopole–antimonopole ordering occurring in a fragmentation state leads to pairing of electric dipoles, so that, in particular, the dielectric response due to these dipoles would be strongly suppressed in the fragmentation state.

In the typical fragmentation state the spins themselves are disordered, which means that such ($\mathbf{d}$, $-\mathbf{d}$) pairs would be random. Each such pair connects two neighbouring triangles, i.e. it lives on the bond of the honeycomb lattice of triangles, and every triangle participates in one such bond. Mathematically this problem could thus be mapped to the problem of dimer covering of a honeycomb lattice—the lattice of triangles in kagome systems. A lot is known about this problem—the entropy of this state, etc.[28,29]. If we go to the long-range ordered state in which the spins are also ordered, this would simultaneously give an ordering of these ($\mathbf{d}$, $-\mathbf{d}$) pairs, i.e. an ordering of such dimers, or of the corresponding electric quadrupoles.

**Pyrochlores**. One can show that the main effects discussed above would also exist in pyrochlore spin ice systems with moment fragmentation. In this case the fragmentation state with monopole ordering would also always contain pairs of ($\mu$, $\bar{\mu}$) with the common 'special' spin (the spin-out for monopole with the (3-in)–(1-out) state and the spin-in for the antimonopole), and correspondingly the dipole moments, pointing to this 'special' spin, would again form ($\mathbf{d}$, $-\mathbf{d}$) pairs, Fig. 2c. Consequently, similar to the kagome case, in a fragmentation state of the bulk system all tetrahedra would always be paired, with the electric dipoles at such pair of tetrahedra pointing opposite to each other, i.e. they would form local ($\mathbf{d}$, $-\mathbf{d}$) pairs, Fig. 2d. Again, these ($\mathbf{d}$, $-\mathbf{d}$) pairs would be random in a fragmentation state with disordered spins, and would be ordered for ordered spins. Mathematically the problem here would be the problem of dimer covering of a diamond lattice (lattice of tetrahedra in pyrochlores). Experimentally one should see these effects e.g. in the reduction of electric activity at the transition of spin ice pyrochlores to a MF state.

**Currents and orbital moments in a moment fragmentation state**. The situation with spontaneous currents and corresponding orbital moments, which may appear at certain magnetic configurations according to Eqs. (2), (3), is considered in details in the Supplementary Information. In short, there appear such currents and orbital moments on monopoles and antimonopoles in pyrochlore spin-ice systems, and also in 'kagome-from-pyrochlores' (or tripod kagome), created by the replacement of triangular [111] layers in pyrochlores by nonmagnetic ions, keeping the kagome [111] layers magnetic[30]. Spin orientation in such systems follows the same rule as in the underlying pyrochlores, i.e. the spins in both systems are noncoplanar, which leads to nontrivial effects connected with currents and orbital moments. The treatment presented in the Supplementary Information shows that the transition to a moment fragmentation state also modifies the pattern of currents and respective orbital moments, in such a way that these orbital moments $\mathbf{L}$ are perfectly ordered in kagome-from-pyrochlore systems, and they are paired (with parallel ($\mathbf{L}$, $\mathbf{L}$) moments) in pyrochlores, but with such pairs remaining disordered. In general, orbital moments are parallel (or antiparallel) to the net spin of respective triangles or tetrahedra, i.e. at first glance they would not lead to any noteworthy physical effects, but there may be interesting effects connected with the scalar spin chirality, see below.

**Defects and excitations in a moment fragmentation state**. Yet another interesting and important aspect to consider is what would be the electric properties of excitations, defects and textures in the moment fragmentation with ordered ($\mu$, $\bar{\mu}$) configurations. As always, when there is an ordered state of some kind, one can form different types of defects: local defects, similar to an inverted spin in an antiferromagnet, or domain walls between

domains, which differ by the interchange of $(\mu, \bar{\mu})$ sublattices. In addition, in contrast to the usual two-sublattice systems such as Néel antiferromagnets, here there may appear yet other types of local excitations: in the kagome MF states with (2-in)–(1-out) configurations, there may appear (3-in) or (3-out) states of a triangle, which would be a novel type of defect. Similarly, such states can appear in pyrochlores: besides ordered $(\mu, \bar{\mu})$ configurations in the MF state, there may exist defects with (4-in) or (4-out) tetrahedra, or the simple spin-ice states (2-in)–(2-out).

Usually such defects are formed in pairs by reversing one spin. By that we exchange magnetic charges at neighbouring sites (by sites we mean triangles in kagome and tetrahedra in pyrochlore systems). Thus when we reverse a 'special' spin in the structures of Fig. 2a, c, we create a pair of 'supermonopoles'—a pair of (all-in)–(all-out) defects, triple monopoles ('tri-poles') in kagome systems of Fig. 2a, with charges $+3Q$ and $-3Q$: and (4-in) and (4-out) states (double monopoles) in pyrochlores, Fig. 2c, with charges $+4Q$ and $-4Q$. (In pyrochlores it is common to define the monopole charge $q$ as $\frac{1}{2}$ of that given above, so that the monopole would have charge $q(=+1)$, and the (4-in) state (double monopole) would have charge $2q$). When we invert not a 'special' but the usual spin in a MF state, we create in kagome a pair of defects with interchanged monopoles and antimonopoles, i.e. with monopoles at the 'wrong' sublattice. In pyrochlores we create by that two usual spin ice states, with (2-in)–(2-out) tetrahedra, with charges 0. In both cases the reversal of one spin changes the charges at adjacent sites (triangles, tetrahedra) by $\pm 2Q$, the total charge being of course conserved. Thus one can say that for example the charge of a tri-pole in kagome case is $3Q$, but also that the excess charge is $2Q$, as compared with the charge $Q$ of the original monopole at this site. Such excitation, when it moves in a crystal, carries with it the excess charge $2Q$. This language is very convenient for considering the motion of such excitations in the MF state.

Once created, these defects can move in a crystal, similar to the case of the usual monopoles in spin ice. The motion of these excitations in a MF state is rather nontrivial. As this question, though definitely interesting, lies somewhat aside from the main topic of this paper—the electric activity of such defects and excitations—I schematically present this discussion in the Supplementary Information; see also[31]. What is important is that such defects can move in a MF state without confinement, similar to the motion of monopoles in the usual spin ice[5], although with certain restrictions. Therefore we can consider the electric activity of isolated defects of this type.

Consider first kagome systems, and start from a defect of the type (3-in) with the charge $3Q$ in place of a monopole (2-in)–(1-out), or (3-out), charge $-3Q$, instead of an antimonopole. This situation is shown in Fig. 3a.

As mentioned above, such 'tri-poles' monopoles with charge $3Q$, or with the excess charge $2Q$, can move in their own sublattice by interchanging some spins, and by moving they leave a trail of changed spins, but such strings have no tension, see Supplementary Information. As to their eventual dipole activity, first one easily sees using Eq. (1) that there would be no charge redistribution and no electric dipoles at such 'tri-pole' triangles themselves. This is also evident just from the symmetry (all three sites and all directions in such a triangle are equivalent). In principle the creation of such a dipole-less triangle (or a defect site in the effective honeycomb lattice of triangles) would remove one dipole from a $(\mathbf{d}, -\mathbf{d})$ pair, so that one could think that there would appear one unpaired dipole. However, by reversing some spins one can in this case 'recommute' the remaining $(\mu, \bar{\mu})$ pairs so that all dipoles would again be paired. And actually one can see that when such tri-pole is one of a pair created at neighbouring

sites (triangles) by reversing a special spin (the tri-pole that moved away from its partner), this pairing of dipoles in between those occurs automatically during the motion of such defects, see Supplementary Information 1. And the string connecting these defects is thus 'decorated' by the reversed $(\mathbf{d}, -\mathbf{d})$ pairs.

The situation might be different for other defects or textures, e.g. created by reversing the usual, not a 'special' spin. The main building block of those is the neighbouring pair of triangles both with monopoles (or antimonopoles). As one sees in Fig. 3b, in this case a common spin is the in-spin for one triangle, but it is the out-spin (i.e. our red 'special' spin) for the other, upper triangle. In effect in this upper triangle the electric dipole would point to this special spin and would not have a 'mate', an opposite dipole at a neighbouring 'site', i.e. it would not form a $(\mathbf{d}, -\mathbf{d})$ pair—it would definitely be an unpaired dipole.

Using this rule it is easy to understand the electric properties of different defects and monopole textures like domain walls. The simplest of those could be the interchange of $\mu, \bar{\mu}$ at neighbouring sites, or just the reversal of a monopole charge at some site, e.g. the replacement of $\bar{\mu}$ by $\mu$ at a particular triangle. The typical resulting situation is shown in Fig. 3c. (One can easily show that such defect can also move in its sublattice without leading to confinement—there would be no string tension on its trajectory either, see Supplementary Information 1.) As to the electric activity of such reversed monopole, one sees that in this case one unavoidably creates unpaired dipoles. A typical building block in this case is formed by three neighbouring triangles all with monopoles (or all with antimonopoles). It is clear that for all possible spin configurations one necessarily has some unpaired dipoles, see Fig. 3c. In effect in a kagome lattice such defects would lead to the creation of even three unpaired dipoles, one at the central 'reversed' triangle, and two dipoles at two out of three adjacent triangles. Thus such defects ($\mu$ at a $\bar{\mu}$ position, or vice versa) would necessarily carry unpaired dipoles.

Actually the situation is very similar on domain walls created by the interchange of $\mu$ and $\bar{\mu}$ in a part of the ordered MF state. There are two types of such domain walls, shown in Fig. 4a, b. At a domain wall of the first type, Fig. 4a, we have a pair of $(\mu, \mu)$ or $(\bar{\mu}, \bar{\mu})$ triangles of the same type, as in Fig. 3b. As is explained there, in this case we necessarily have an unpaired dipole moment perpendicular to such domain wall, Fig. 4a. These dipoles can be random, up or down. The situation is similar for the domain wall of the second type, Fig. 4b, except that such neighbouring $(\mu, \mu)$ triangles would be oriented differently—more or less along the domain wall (in a tilted zigzag pattern). Consequently also in this case, by the same reason, there would appear unpaired dipoles along such domain wall, Fig. 4b.

In fact the situation is rather similar for defects and spin textures in pyrochlores with moment fragmentation. By the same reason— partial ordering (ordering of monopoles with still free spins), local defects are, first of all, mobile, i.e. it does not cost a linearly increasing energy to move them in their own sublattice—due to the freedom to reverse spins on the MF state there would be no confinement in this case. Defects of the (4-in) (or (4-out)) type would again have features of monopoles with a larger magnetic charge: monopoles with (3-in)–(1-out) state have magnetic charge $+2Q$, and the (4-in) state has charge $+4Q$, i.e. in this case the excess charge of the excitation itself is also $2Q$.

As to dipole activity, one can again see that defects of this type—(4-in) tetrahedra at a $\mu$-site or (4-out) at a $\bar{\mu}$ site of the MF state should not necessarily create any dipoles: such sites (tetrahedra) themselves are symmetric and do not have any dipoles; and one can always recommute spins on neighbouring tetrahedra to get rid of eventual unpaired dipoles. And again, when we first create a pair of such double monopoles

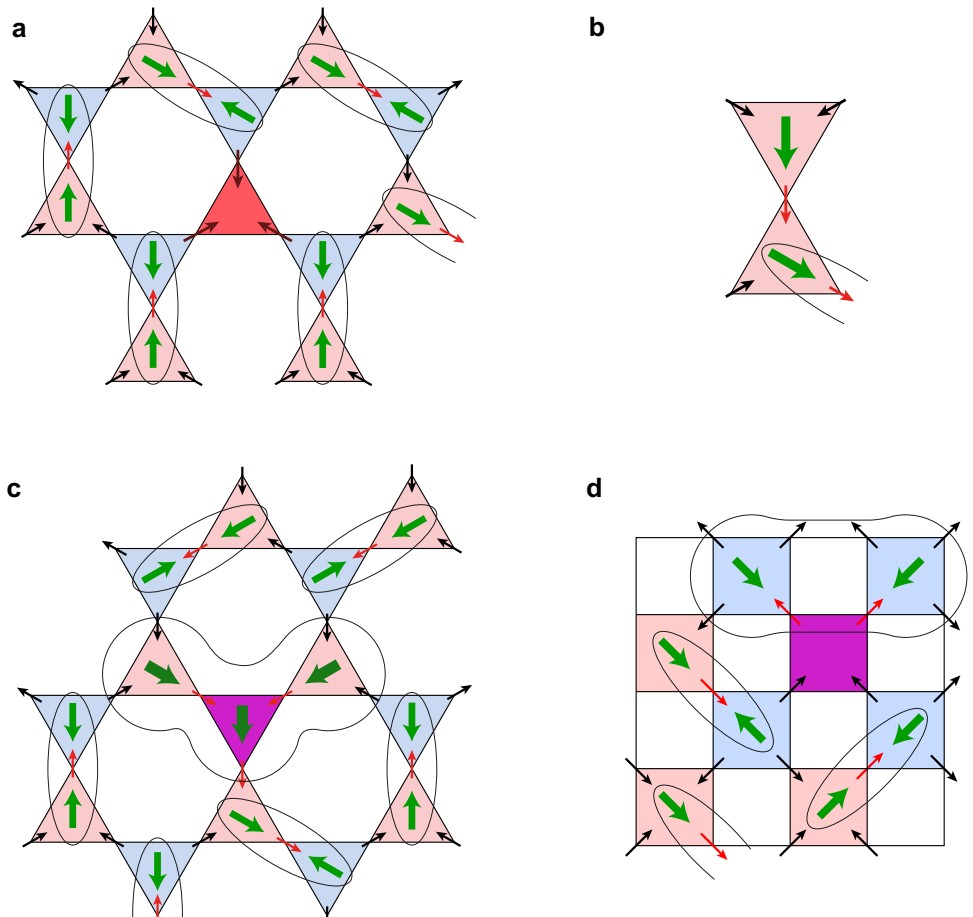

**Fig. 3 Dipole structure at defects and monopole textures in kagome moment fragmentation state. a** Triangle with (3-in) state ('tri-pole'). Reconnection of free spins allows to create such defect without free dipoles. **b** Typical situation with the monopole at the 'wrong' place, at the antimonopole sublattice, demonstrating the appearance of unpaired dipoles. Note that the common spin is the 'normal' in-spin for one triangle, but it is a 'special' out-spin for the other triangle. Consequently the dipole moment at the second, upper triangle would point to this 'special' spin and would be definitely unpaired. **c** Monopole in place of an antimonopole (magenta triangle). One sees that it inevitably leads to the creation of three unpaired dipoles: on the defect triangle itself and on two (out of three) of its neighbours. **d** The appearance of (two) unpaired dipoles next to (2-in)--(2-out) tetrahedron (magenta) in pyrochlore with moment fragmentation.

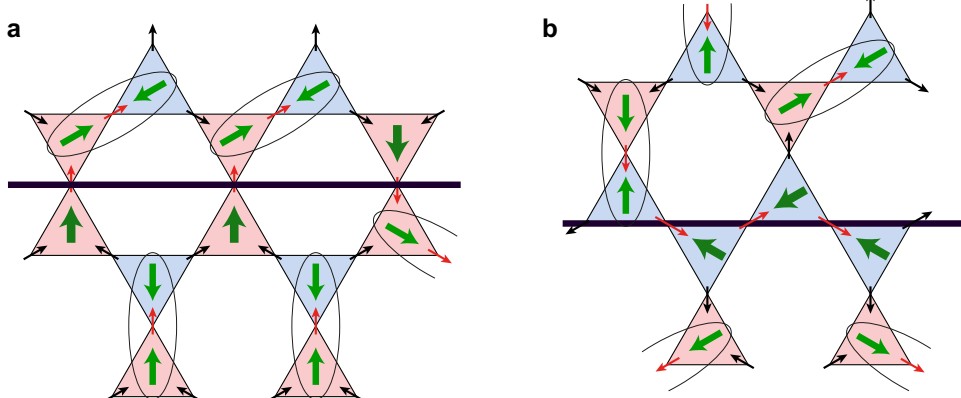

**Fig. 4 Dipole structure of domain walls in kagome systems with moment fragmentation. a, b** Monopole domain walls of type 1 and 2. There are unpaired dipoles on domain walls of both types. The situation would be in principle similar for pyrochlore systems with moment fragmentation (shown in Fig. S4 of the Supplementary Information).

with charges $+4Q$ and $-4Q$ by reversing the special spin in Fig. 2c (effectively destroying the dipoles of the $(\mathbf{d}, -\mathbf{d})$ dimer on these neighbouring sites), then moving these defects apart we automatically recommute spins in such a way as to create

on the trajectory of these defects new $(\mathbf{d}, -\mathbf{d})$ pairs, see the Supplementary Information.

On the other hand, the 'non-monopole' spin ice (2-in)–(2-out) defects, created e.g. by reversing the usual spins (also mobile

without confinement) would again, similar to the kagome case, create an unpaired dipole (actually two of them), see Fig. 3d. As is clear from this figure, two neighbouring tetrahedra adjacent to the (2-in)–(2-out) defect (which in itself has no electric dipole) should necessarily have unpaired dipoles directed towards such defect.

For other types of defects or textures, e.g. for domain walls, the main 'building block' is two neighbouring tetrahedra with one common site both having monopoles (or both antimonopoles). Similar to the situation of Fig. 3b, this pair also necessarily has at least one unpaired dipole directed towards the 'special' spin of the common site, the dipole at neighbouring tetrahedron pointing away from it, so that at least the first dipole cannot have a 'mate' with opposite direction. Therefore the situation in monopole domain walls for pyrochlores is also similar to that in kagome systems. In this case there would also be present nn pairs of tetrahedra both with monopoles (of both with antimonopoles), and by the same reason as above there should necessarily exist unpaired dipoles at such domain walls. The details of their orientation are described in the Supplementary Information. The interchange of spins inside every domain can lead to reversal of some such dipoles, so that generically the monopole domain walls in the MF state should have extra electric activity, both in kagome and in pyrochlore spin ice systems with MF.

## Discussion
In this paper we demonstrated that in spin ice systems (kagome, pyrochlores) there exist quite nontrivial electric properties when there occurs in them a moment fragmentation—the formation of a partially ordered state in which monopoles and antimonopoles are present on every triangle in kagome and every tetrahedron in pyrochlores and are ordered in a two-sublattice fashion, while spins themselves are still disordered. This partial ordering strongly influences electric dipoles attached to every monopole and anti-monopole in both these systems. Whereas without moment fragmentation these dipoles are random and dynamic (fluctuating together with spins), in the fragmentation state they are paired into $(\mathbf{d}, -\mathbf{d})$ pairs on neighbouring triangles or tetrahedra. Therefore, the transition to a fragmentation state should lead to a reduction of electric activity present in spin ice systems without fragmentation due to random and dynamic dipoles, which is seen e.g. in microwave absorption[13].

On the other hand, electric currents existing in triangles with noncoplanar spins[25,26] behave differently in kagome and in pyrochlore systems. In kagome systems ('kagome-from-pyrochlores', see Supplementary Information) such currents and the orbital moments associated with them, random in the usual case, become long-range ordered simultaneously with the ordering of monopoles at moment fragmentation. In pyrochlores, however, such currents and the corresponding orbital moments in the fragmentation state remain random, but are also paired, but in this case into $(\mathbf{L}, \mathbf{L})$ pairs.

The formation of staggered current pattern and staggered scalar spin chirality in 'kagome-from-pyrochlores' may lead to quite nontrivial effects. It is known that the existence of scalar spin chirality and the corresponding Berry phase leads to the appearance of very strong fictitious magnetic field. For total nonzero spin chirality it leads, in particular, to the intrinsic anomalous Hall effect[27].

Very interesting and at first glance surprising is the electric behaviour of different defects in a MF state. Such point defects or excitations, like the 'supermonopoles' ((all-in) and (all-out) states) can in principle be generated by external disorder, impurities etc., but they can also be excited thermally with the excitation energy different from system to system but typically of the order of the transition/crossover temperature to the MF state.

Thus e.g. in $Dy_2Ir_2O_7$ this excitation energy is ~ 3.6 K, whereas the crossover to the MF state occurs at ~ 1.4 K[21]. Probably one can also create such excitations by external perturbations, e.g. by microwave irradiation.

In contrast to naive expectations not all such defects break $(\mathbf{d}, -\mathbf{d})$ pairs and lead to a creation of unpaired dipoles. The defects of monopole type with larger magnetic charge ('tri-poles' in kagome—(3-in) or (3-out) triangles) and 'tetrapoles' in pyrochlores ((4-in) or (4-out) tetrahedra) do not in general lead to unpaired dipoles. However, other defects or textures—e.g. (2-in)–(2-out) states in pyrochlores and monopoles on 'wrong' anti-monopole sites—breaks the $(\mathbf{d}, -\mathbf{d})$ pairs and necessarily lead to the creation of unpaired dipoles. The same is true for domain walls in the $(\mu, \bar{\mu})$ ordered state in both kagome and pyrochlore systems: these also carry unpaired dipoles. Thus the creation of such defects or textures generically should be accompanied by an increase of electric activity of moment fragmentation systems.

Such defects in the partially-ordered fragmentation state have their own interesting dynamics—typically they can be moved without confinement, similar to monopoles in the usual spin ice. And at high concentration such defects would start to interact with each other, in particular by the dipole-dipole interaction, which can lead to rather nontrivial behavior[10].

The study of defects and their dynamics, both magnetic and electric, in the moment fragmentation state is yet experimentally a completely unexplored field, which definitely deserves the interest of specialists.

Let us now consider some possible experimental manifestations of the effects discussed above.

1. The main effect is the pairing of electric dipoles in a MF state into $(\mathbf{d}, -\mathbf{d})$ pairs, both in pyrochlore and in kagome spin ice. The most apparent consequence is the corresponding reduction of electric activity in this state, e.g. measured by microwave absorption, as used e.g. in[13]. In pyrochlores the most reliable observation of a MF state was made in Ir pyrochlores such as $Dy_2IrO_7$, in which the (all-in)–(all-out) ordering of Ir at high temperatures gives staggered field for monopole sublattices[20], so that in this case the transition to a MF is not a sharp transition but rather a crossover with broad anomalies[21]. In principle the effects could be more prominent in systems in which the MF state would appear as a real well-defined transition. However, e.g. the results claiming such transition in $Nd_2Zr_2O_7$[17] were recently questioned[32], and it seems that such transition without external field is difficult to realise in pyrochlores. The situation in this sense may be better in kagome ice systems. There are reliable reports that such transition was indeed observed in these systems, both natural and artificial[18,30]. The existence of such well-defined MF phase 'sandwiched' between the disordered high-temperature phase and the fully ordered phase at low temperatures was also obtained theoretically[18,22,23]. Kagome systems may be preferable also from another point of view. In pyrochlores the monopoles are the excited states, thus to observe the effects discussed above (such as the suppression of electric activity when going to the MF state) one first has to create monopoles with attached electric dipoles, e.g. by changing the temperature, magnetic field etc., which would increase the electric activity, and then to order the monopoles in a MF state, forming $(\mathbf{d}, -\mathbf{d})$ pairs, which would lead to a reduction of this activity. In kagome ice systems, on the other hand, monopoles with their dipoles exist in all phases, and the reduction of electric activity in going to a MF state would be seen more clearly.

The formation of dipoles at monopoles in spin ice is associated with the corresponding shifts of oxygens sitting in the middle of metal tetrahedra; one sees these shifts e.g. by Raman scattering[14]. The transition to a MF state would then lead, together with the formation of $(\mathbf{d}, -\mathbf{d})$ pairs, to correlation of these shifts at adjacent tetrahedra or triangles. These correlated distortions could

probably be also investigated by Raman scattering or by some other technique.

2. But probably even more interesting would be the experimental situation for monopole textures and defects, such as monopole domain walls. The unique feature of a MF state is the coexistence in one spin subsystem of an ordered and a disordered components. As always, each time we have long-range ordering in a system, the most interesting effects are connected with excitations violating this order[33]. However, just for a MF state, with its remaining spin disorder, such DW would be rather elusive, there would be no prominent magnetic contrast on them. Probably just because of that there was no study, even no mention of such DW in the literature. And the fact that, as shown above, such DW (and also many point defects) would be 'decorated' by electric dipoles, would make their observation possible, for example by STM, sensitive to local electric fields thus created (e.g. using electron standing waves[34]). And the electric activity of such DW could in principle allow one to influence and control them by electric field, as was done e.g. for ordinary DW in ferromagnets in[35]. Similarly one could think of controlling point defects with uncompensated dipoles by an inhomogeneous electric field, cf. the study of skyrmions by such method in[36].

3. As is shown above, in particular in SI, the MF state should also display nontrivial patterns of scalar spin chirality and of spontaneous currents and respective orbital moments. Somewhat unluckily, in all cases considered the direction of these orbital moments $\mathbf{L}$ for every tetrahedron or triangle in pyrochlore and kagome systems turned out to be parallel to the respective net spin of these, $\mathbf{S}_{tot}$, and typically the orbital moments would make only a small fraction of the full moment $\mathbf{M} = \mathbf{S}_{tot} + \mathbf{L}$. Thus, I don't expect interesting purely magnetic effects connected with those. But on the other hand we get here a nonzero scalar spin chirality $\kappa$, Eq. (3), the same for every triangle in kagome-from-pyrochlore systems, see Fig. S6. Similarly, such 'ferro' scalar chirality would exist in kagome ice state, created in spin ice pyrochlores in the [111] magnetic field slightly below the transition to the fully-ordered state, see e.g.[24]. And one should expect that this scalar spin chirality would give the intrinsic (Berry phase) anomalous Hall effect, similar to the case of $Nd_2Mo_2O_7$[27]. This effect would be very interesting to check experimentally.

In conclusion we want to stress once again a very special and unusual feature of moment fragmentation state in spin ice: the coexistence in one spin system of one component (monopoles, a divergence-full part of magnetization) with perfect long range ordering, and a disordered component (divergence-free part of magnetization). This strange situation with partial ordering leads to many unusual properties, such as the existence of novel types of defects and excitations, domain walls with special properties, etc. And we demonstrated that the electric activity of this state is also very special: on one hand, due to ordering in the monopole–antimonopole sector the electric dipoles are paired into $(\mathbf{d}, -\mathbf{d})$ pairs, and, on the other hand, due to spin disorder these dipole pairs are random and dynamic. The same dichotomy of the moment fragmentation state is also reflected in the electric properties of defects and domain walls. This two-faced Janus character of this state makes these systems especially interesting and unusual.

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

## Acknowledgements
I am very grateful to L. Chapon and M. Mourigal for introducing me to the field of moment fragmentation in spin ice, and to T. Michely and G. Jackeli for useful discussions. This work was funded by the Deutsche Forschungsgemeinschaft (DFG, German Research Foundation)—Project number 277146847—CRC 1238.

## Author contributions
D.I.K. is responsible for the whole content of this paper.

## Funding

## Competing interests
The author declares no competing interests.
