## [Peer Review File · Nature Communications]

Reviewers' Comments:

Reviewer #1:

Remarks to the Author:

The phenomenon of magnetic moment fragmentation has been proposed and recently observed in materials closely related to spin ice systems. It occurs when a crystal of "magnetic monopoles" (quasiparticles arising in Ising pyrochlores under certain conditions) can be stabilized in such a way that spin fluctuations are not totally frozen. Besides its magnetic charge and magnetic moment, it has been demonstrated that each monopole has necessarily an associated dipolar electric moment. In this work the author studies what happens with these electric dipoles, and also to electric currents and orbital moments, when a system undergoes magnetic moment fragmentation.

The manuscript has two main parts. The main result is presented in the first one: it is shown that the advent of single monopole order in a crystal with fluctuating magnetic moments is accompanied by a pairing of electric dipoles, which results in a reduction of electric activity. There is no quantitative estimation of this reduction, but it is claimed that it should be observable in microwave absorption (used in previous experiments to detect electric dipolar effects near a transition in which monopoles abruptly condense, ref. [9] in the text). The second part of the manuscript deals with the electric properties of defects in the fragmented monopole crystal.

The work is original, and represents an interesting contribution to the knowledge of the interplay between magnetic and electric effects on "monopole matter", and the possibility to control these materials in a multiferroic-like fashion. Although it is mainly qualitative, it is bound to have impact within the theoretical and experimental community of geometrically frustrated magnetic systems. However, I would like the author to clarify some points before I can make a recommendation.

* The prediction of a pairing of electric dipoles in the fragmented phase, in the first part of the text, is a very clear advance. It would benefit from a more explicit prediction of how it could be detected. For example, since the symmetry between up and down tetrahedra is broken at high temperatures in Dy₂Ir₂O₇, the monopole crystal is stabilized rather gradually (there is only a broad peak in specific heat; see for example Cathelin, V., et al, *Physical Review Research*, 2(3), 032073 (2020)). How can one be sure to detect the effect studied here in this or other compounds?

I find the second part of the manuscript somewhat vaguer than the first one, in particular regarding the physical basis where it stands.

* Different cases are analysed (for example, fully paired or unpaired dipoles for an isolated triple monopole in the Kagome lattice, domain walls, etc.) without providing an estimation for their importance at a given temperature, or their probability of occurrence taking into account their energy cost or associated entropy gain. As far as I can tell, the author does not provide nor mention the underlying Hamiltonian.

* Dipolar interactions are usually quite important in these systems (see for example Cathelin, V., et al, *Physical Review Research*, 2(3), 032073 (2020)). They are likely to have much influence on the presence and spacial correlations between these defects. The author should explain or justify not taking into account these effects.

* When evaluating the "tension" of strings of spin: it would be useful to be specific about the value

of the charge at attached at each end of it, and also of the implicit Hamiltonian used to estimate it.

* The author says that "(...) in real situations the creation of such (3-in) or (3-out) triangles most probably would be accompanied by the appearance of some unpaired dipoles, just because the recombining of other (μ - μ) sites could have relatively slow kinetics, i.e. their creation could still lead to some increase of electric activity of the material." Written in this way, it seems to be implicit that the system is spontaneously driven into a state of full dipole pairing. However, there is no mention of the mechanisms by which this would occur.

I list below some other minor points.:

- I find the use of the term "tripole" for a charge $3Q$ somewhat misleading, in particular if compared with the common use of "dipole" (which implies spacial charge distribution more than the value of the total charge). May be the use of "triple monopole" (like the "double monopoles" within the spin ice context) would be better?

- In the Introduction, it is said that: "Especially interesting is the state in which magnetic monopoles and antimonopoles exist on each metal tetrahedra in pyrochlores and are ordered in a two-sublattice fashion (the tetrahedra in pyrochlores and at B-sites of spinels form a bipartite diamond lattice)."

What are these "metal tetrahedra"?

- As far as I can tell, μ -triangles are not defined in the main text.

- There is probably something wrong in the spin configuration of the blue triangle in Fig. 2a).

Reviewer #2:

Remarks to the Author:

This paper studies the distribution of electric dipoles induced by magnetic charge in spin ice and kagome spin ice models. The author considers dipole distributions in states with magnetic charge order, in which the emergent magnetostatic field of the spin distribution fragments into two parts via a lattice Helmholtz decomposition. In the case of perfect magnetic charge order, the dipoles are paired into a quadrupolar liquid (with dipolar correlations !). Defects to the charge ordered state lead to free and disordered dipoles.

The paper contains an interesting observation. However, it is extremely qualitative and contains a number of imprecise statements. In consequence, I do not see enough science to warrant publication in Nature Communications as the paper stands. Points to consider are detailed below.

1. At present, the paper contains little more than the observation of pairing of electric dipoles into quadrupolar units and their possible breakup through excitations, while a more detailed analysis related to symmetry and/or to the Helmholtz decomposition of fragmentation seems possible. The author states, on page 8 "The first guess could be that there would appear unpaired electric dipoles at all such defects..." This was not my first guess – the generation of an electric dipole clearly comes from the reduction in symmetry of the magnetic configuration on a vertex compared with the point group symmetry of the vertex itself. As all in or all out spin configurations maintain this symmetry how can such a vertex generate an electric dipole ? In fact, this argument is repeated further down on page 9 and is a consequence of the resumé of refs [20,21] on page 4.

As a consequence there should be no room for guessing here. It may be just a question of presentation but this part of the paper could be greatly improved.

2. Similarly, the discussion of defect populations is imprecise. On page 9 the author writes « Thus in principle such "tripole" defects or excitations do not necessarily lead to the formation of free unpaired dipoles " and "(or, more exactly, the unpaired dipole would be pushed to the edge of the sample)". Is this correct ? Both unpaired dipole defects and all in or all out defects are deconfined and both have a finite energy scale for creation, so I would have expected to have a fluid of these two objects in the bulk of the sample with their densities controlled by their respective Boltzmann weights. Again, this discussion should be tightened up, or my remarks should be refuted.

3. The author further discusses defect mobility ""First, one may show that such "tripoles" (monopoles with charge $3Q$) are mobile in MF kagome systems ". It may again be a question of language but I do not think this statement is correct. Following any path taken by a defect, either between sites of one sublattice so that localized charge is $3Q$, or by nearest neighbor hopping so that on alternate steps one visits sites with $Q, 3Q, Q \dots$, the mobile charge is $2Q$. That is, it is the charge of an emergent monopole in the language of reference [4]. Again, the authors should rectify or refute this point. The authors comment on differences in monopole nomenclature in the literature. This is related to this point – it is the difference between the "bookkeeping charge" of a multipole expansion and the charge on emergent excitations, as discussed in journals.aps.org/rmp/abstract/10.1103/RevModPhys.85.1473 in the context of artificial spin ice.

4. It is certainly interesting that scalar defects in the magnetic sector, either bookkeeping or emergent, induce vector defects in the electric sector. Is it possible to make a more systematic analysis of how these vectors are created and annihilated including what quantities are conserved and how they interact ? Are they topological, do they have their own emergence associated with them independently of the magnetic charge ? Quantitative facts here would again make the paper much stronger than the present text and would relate to the interesting discussion of domain walls already present.

5. The fact that a charge – anticharge pair in the magnetic sector generates a pair of anti-aligned electric dipoles is interesting. It would be nice to have a more detailed discussion of this absence of time reversal symmetry and where it comes from, even though this may touch on work from previous papers. It would also be interesting to have some insight into the role of fragmentation in this regard.

Minor points:

1. There is an error in the blue, down triangle of Fig. 2a – all three spins are pointing in.

2. Concerning your ref. [12] – although ref. [11] does deal principally with the monopole and charge crystal phases, the paper makes it quite clear that fragmentation applies to any monopole configuration, not just those with broken symmetry in the charge configurations.

Peter Holdsworth, Lyon, 06/11/2020

Reviewer #3:

Remarks to the Author:

In this manuscript, the author considers the electrical properties of a class of frustrated magnets which exhibit the phenomenon of "moment fragmentation". Moment fragmented systems exhibit antiferromagnetic order coexisting with a large number of fluctuating, algebraically correlated degrees of freedom. This phenomenon has been shown to occur in real materials in recent years, which has generated considerable interest in the topic.

This new work considers the effect of moment fragmentation on the electrical properties of the system. Specifically, the author considers the electric polarisation and current loops which are allowed to exist even in a Mott insulator because of the ability of charges to slightly delocalize over neighbouring sites.

This is an interesting aspect of the problem to consider, since it is known that the monopoles which crystallize to form the moment fragmented state carry electric dipoles.

The author shows that upon entering the moment fragmented state these dipoles pair up into opposing pairs, reducing the electrical fluctuations. The author further show that different flavours of excitation above the ground state have different electrical properties and that domain walls in the moment fragmented state carry electrical polarization, opening up the possibility that they can be controlled with applied electric fields.

I found the paper interesting, well written and the analysis convincing.

Moment fragmented systems are of great interest at the moment, and the coupling between magnetic and electric degrees of freedom in strongly correlated systems is an important topic in its own right. I therefore think this work will be of significant interest to the field and will encourage further investigations, both experimental and theoretical, into spontaneous polarization and orbital currents in related frustrated systems.

I recommend this manuscript for publication in Nature Communications.

I have one minor comment for the author to consider:

- When describing the experimental evidence for moment fragmentation the author includes Ref. 13, which is a study of the material Nd₂Zr₂O₇. Although Nd₂Zr₂O₇ indeed realizes a kind of moment fragmentation, it is not really the same as the kind discussed in this manuscript, because in that material it is really an effect to do with the spin dynamics, not the ground state, although this was not understood when Ref. 13 was written. [For a discussion of this see: O. Benton, Phys. Rev. B 94, 104430 (2016); E. Lhotel et al, J. Low. Temp. Phys. (2020)].

I think the type of moment fragmentation discussed by the author is best exemplified by the kagome systems, and Ho₂Ir₂O₇. I am not sure the same discussion would apply to Nd₂Zr₂O₇.

Typos:

- Page 5: deterred -> deferred

Response to the report of Referee 1

I am grateful to the Referee for his positive recommendations, but even more so for several useful remarks made, which forced me to rethink some places and some formulations in my text. I hope I successfully answered below the concerns of the Referee, and I followed several of his/her recommendations.

Actually a large part of remarks of all Referees concern one particular part of my paper and are related to the properties of defects and excitations – their charge, mobility etc. These questions are indeed important, although they are not directly related to the main topic of my paper – the electric activity, electric dipoles at these defects. Still, probably this part was not explained clearly enough in my text. Therefore I now made several modifications in the presentation of this material in the main text, in the section “Defects and excitations in a moment fragmentation state”, and I also introduced a special large part in the Supplementary Material, section SI 1, in which I discuss in details the creation and motion of these defects and excitations. I hope these modifications give sufficient clarifications and answer several questions/comments of the Referee. All modifications made in the text are marked by blue colour (unfortunately not well seen in the printed copy in the electric light, but clearly seen in daylight and also on the computer screen).

Now to the specific remarks of the Referee. The first recommendation of the Referee 1 is that the paper “*would benefit from a more explicit prediction of how it (the effect predicted) could be detected*”. As the Referee mentions, there indeed may exist two situations. In one, in systems such as $\text{Dy}_2\text{Ti}_2\text{O}_7$ or in some kagome systems the appearance of the ordered state with moment fragmentation (MF) at low temperatures reduces the symmetry of the system and would be a real phase transition in the Landau sense. In this case the anomalies in electric response, e.g. in microwave absorption, due to the change of dipole pattern accompanying MF, should have sharp features at T_c , similar to those observed in [9]. Unfortunately this usually occurs below 1 K and is not so easy to detect. On the other hand in pyrochlores such as Dy_2IrO_7 the (all-in)–(all-out) ordering or Ir at high temperatures gives staggered field for monopole sublattices [17], so that in this case the MF is rather a crossover with broad anomalies [Cathelin] (I am grateful to the Referee for pointing out this very useful reference, which I have now added to the paper, new Ref [18]). In this case, in contrast, this crossover occurs in a more accessible temperature range $\sim 1\text{--}4$ K. In both cases the appearance of MF state would lead, as is shown in the paper, to pairing of dipoles in $(\mathbf{d}, -\mathbf{d})$ pairs and to a corresponding weakening of the electric response, to the reduction of microwave absorption. It also opens the possibility to influence/control especially the defects and excitations in this state by electric field. I now introduced the corresponding discussion in the text.

The second remark of Referee 1 is that the discussion of different defects in MF state and their associated electric activity would benefit from a more detailed discussion of specific conditions, e.g. at which temperatures they may be relevant and observable. This is indeed a valid and important point. Some of these defects are in fact excitations in a MF state, with specific activation energy. This activation energy depends on a particular system. In general it is of the order of the temperature of corresponding ordering. Thus, by the data of [18], in $\text{Dy}_2\text{Ir}_2\text{O}_7$ the excitation energy of a “supermonopole” – (4-in) or (4-out) state of a tetrahedron, is ~ 3.6 K, the crossover temperature of the formation of MF state being 1.4 K. The others textures, like domain walls, may depend on the particular features of a given sample, on the distribution of defects in those with eventual pinning of domain walls, on the thermal history of measurements. Thus it is difficult to estimate the corresponding conditions for their appearance and their eventual temperature dependence. To clarify all these points I have now added in my text the corresponding discussion.

The next question of Referee 1 concerns the possible role of dipole–dipole interaction. It is not quite clear whether the Referee has in mind magnetic dipole–dipole interaction or the interaction of electric dipoles appearing on monopoles in spin ice. As to the interaction between magnetic dipoles, it is well known to play a very important role in the very phenomenon of spin ice; in this sense my study does not add or change anything. But when we have monopoles and antimonopoles in a system, there would appear on those also electric dipoles [8], which, as demonstrated in the present paper, would be paired in $(\mathbf{d}, -\mathbf{d})$ pairs in a MF state. There should be of course also a dipole–dipole interaction between such electric dipoles. But in a MF state these dipoles are paired in effective quadrupolar objects, therefore just in a MF state the role of these interactions would be strongly reduced. However when one creates defects or excitations with unpaired dipoles, their dipole–dipole interaction with other such defects may become important. But in my paper I considered only the properties of isolated defects of this type; I have not considered the situation with high concentration of such excitation. For high concentration of those the dipole–dipole interaction might indeed become important, but the treatment of such highly-excited state, with its dynamics, lies far beyond the scope of my paper. My task was to investigate in which situation, for which defects, free dipoles would appear and how big would they be, what would be their orientation, etc. This is the basic information, the basis which later on could be used to treat the much more complicated case of strongly excited MF state.

The next point raised by the Referee concerns the tension of the string created when defects are moving on MF background, and charges at the ends of such strings. Actually the main point I wanted to make in mentioning these strings is just that here, in a MF state, in contrast to most systems with long-range ordered ground state, due to the presence of still disordered spins there would be *no* tension of such strings (but not all trajectories of the defects satisfy this condition, as is explained in the new section SI 1 of the Supplementary Material). In this sense, despite the existence of ordering of monopoles, the situation is rather similar to that for monopoles and respective strings in the usual spin ice [4]. I only wanted to stress here the unique feature of the MF state: that despite long-range ordering of monopoles there is no tension of strings, the energy increasing linearly with their length, and consequently there is

no confinement of such excitations. As to the magnetic charges at the ends of these strings, this question I now also discuss in detail, in the main text but mainly in this new part SI 1 of SI. In short, the excitations carry constant excess charge – e.g. the charge of a triple monopole instead of a monopole, (3-in) state in kagome system, is $3Q$, but its excess charge is $(3Q - Q) = 2Q$, and this excess charge is conserved and moves in a crystal. And moving it creates again a tri-pole at the monopole sublattice, with the total charge $3Q$ instead of the original charge Q at this site, but at the next step, when the excitation is on the other sublattice, it will be a “monopole in a wrong sublattice”, with the charge $(-Q + 2Q) = +Q$. All these points are now discussed in details especially in the new section SI 1 of the Supplementary Material.

The last remark of the Referee 1 is that the sentence in my text that “*in real situations the creation of such (3-in) or (3-out) triangles most probably would be accompanied by the appearance of some unpaired dipoles, just because the recommitting of other spins at sites could have relatively slow kinetics, i.e. their creation could still lead to some increase of electric activity of the material*” is somewhat imprecise and ambiguous. I tend to agree with this criticism. This sentence is actually physically correct but it is just a qualitative statement, I cannot really substantiate it. It is probably better to avoid making imprecise statements like this. Therefore I decided to omit this sentence – although physically it is most probably correct.

As to some minor points mentioned in the report of Referee 1:

1. The term “tripole” may be indeed not very lucky. I changed it to “triple monopole” as suggested by the Referee, and in some places for shortness I write “tri-poles”, to avoid direct resemblance with “dipoles”.
2. As to “metal tetrahedra” in pyrochlores: I think the meaning of this term is quite evident. Metal ions, in particular rare earth ions responsible for the spin-ice behaviour, form corner-sharing tetrahedra in pyrochlore lattice; I think this is clear.
3. I changed the wording in the part of the text where I originally mentioned “ μ -triangles”, as the Referee suggested.
4. I myself have already noticed this mistake in spin configuration in Fig.2(a); in the version of the paper which I put on the arXiv server it was already corrected.

Altogether I am grateful to Referee 1 for careful reading of the manuscript and for many useful comments and suggestions, which, I hope, helped me to improve the presentation.

Response to the report of Referee 2

In contrast to Referees 1 and 3, Referee 2 disclosed his identity. I am actually very glad that this Referee was Prof. P. Holdsworth – one of the “fathers” of this whole field of moment fragmentation in spin ice, apparently the main author of the main PRX paper [11] which opened this whole field. His opinion is very important to me. He is more critical of the paper, although in fact he seems to agree with my main conclusions about electric activity in a MF state and with the picture emerging from my paper. His main complaint is that in his opinion my paper is very qualitative. I may even agree with him as to this point. But the question is whether the qualitative character of the paper is a negative or rather a positive characteristic, a drawback or a virtue of the paper. In my opinion if the paper contains simple, novel but easy-to-understand statements of qualitative character, it makes the paper more accessible and may only increase its impact. The main statements of the paper may sometimes even look almost self-evident, but only *after* they have been made. My paper is based on the solid mathematical framework, described in Eqs. (1)–(3). But indeed the main effects discussed in the paper may be understood on a qualitative level. I don’t think this is a drawback of the paper. In a sense the main message of the first main paper by the Referee, Ref. [11], may also be formulated as a qualitative statement in one or two sentences: that “by using Helmholtz method one can decompose the magnetization of, in particular, spin ice systems into two components, a divergence-free part (random spins) and a divergence-full part (monopoles), and the second part may even become perfectly ordered, whereas spins themselves remain disordered”. Of course there is much more material in the paper [11], but I think that just this simple and easy-to-grasp statement attracted so much attention to this problem and has led to a flurry of experimental activity following this suggestion. I suspect that probably not many people, especially experimentalists, looked much further in the paper [11] beyond this main result. Thus in this sense the very fact that the paper [11] contained very important but easy-to-understand qualitative result, was a strong, not a weak, point of this paper. I hope something similar, to a lesser extent, may happen with my paper.

Referee 2 also asked a few specific questions, which I address below. Some of these remarks are rather deep, but some deal with a somewhat imprecise use of some terms in my paper. Below I will try to clarify these points, and in many cases I have made corresponding changes in my manuscript.

Actually the main remarks of the Referee, as well as some remarks of Referee 1, concern one particular part of my paper and are related to the properties of defects and excitations – their creation, charge, mobility etc. These questions are indeed very important, although they are not directly related to the main topic of my paper – the electric activity, electric dipoles at these defects. Still, probably this part was not explained clearly enough in my text. Therefore I have now made several modifications to the presentation of this material in the main text, in the section “Defects and excitations in a moment fragmentation state”, and I also added a special large part in the Supplementary Material, section SI 1, in which I discuss in details the

creation and motion of these defects and excitations. I hope these modifications give sufficient clarifications and answer the main questions/comments by Referees.

Now to the detailed remarks of Referee 2.

1. The first remark concerns the sentence in my paper that “The first guess could be that there would appear unpaired electric dipoles at all such defects”. The Referee writes that “*This was not my first guess; the generation of an electric dipole clearly comes from the reduction in symmetry of the magnetic configuration on a vertex compared with the point group symmetry of the vertex itself. As all in or all out spin configurations maintain this symmetry how can such a vertex generate an electric dipole?*”. My arguments were the following: As I have shown in this paper, dipoles are paired in a moment fragmentation (MF) state in $(\mathbf{d}, -\mathbf{d})$ pairs. And when I remove one dipole of such a pair, e.g. making one tetrahedron in pyrochlore or one triangle in kagome a higher multipole, all-in state, this defect itself of course does not have a dipole, but its “mate”, the dipole which was initially paired to it, would remain without a pair and would be a free dipole. This is actually very similar to the situation with the creation of spinons in RVB systems: when we replace one site of a singlet dimer by a nonmagnetic impurity, the remaining site would have an unpaired spin. Exactly the same could have happened in this case; this was the meaning of my observation “The first guess could be that there would appear unpaired electric dipoles at all such defects”. And only because of the possibility to recombine the remaining spins in some cases – in particular in this all-in defect – one can remove this remaining unpaired dipole. But indeed, the way this was presented in my paper, could have raised such doubts. Therefore I have now changed the formulation in this place. And especially with the new presentation of this topic, in the main text and in the new part SI 1, I hope this point is now clarified.

2. In this remark the Referee writes that “*the discussion of defect populations is imprecise*” and that “*Both unpaired dipole defects and all in or all out defects are deconfined and both have a finite energy scale for creation, so I would have expected to have a fluid of these two objects in the bulk of the sample with their densities controlled by their respective Boltzmann weights*”. Of course the Referee is completely right at this point, and I have not said anything in my paper contradicting this. I just did not consider the situation with many such excitations; my aim was to investigate the electric activity, dipole character of different such isolated defects. Of course if one would consider the situation at finite temperatures, the thermodynamic and dynamic properties of systems with finite concentration of such excitations, one should have done what the Referee writes. But, once again, my aim was quite different – it was the question which defects or excitations would have electric activity and how the dipole structure of those would look like. It is a completely different task, a different question than what the Referee mention in this remark – with which, once again, I completely agree.

3. The third remark of the Referee is actually very important and useful; it concerns the question whether, when moving e.g. the “tri-pole” (all-in triangle) in a MF state of kagome lattice, one should speak about the motion of an object with magnetic charge $3Q$, which this “tri-pole” itself has, or whether one should rather speak about the “excess” charge: this tri-

pole with charge $3Q$ replaces the monopole with charge Q and can move without confinement, so that the “extra” charge which moves through the crystal is $3Q - Q = 2Q$. The Referee prefers to use the second definition and speak about the “excess” charge. In a sense this is a matter of convenience; important is only that we understand what we mean. And indeed, for some questions it is more convenient to speak about the excess charge which the excitation carries when it moves through the crystal, i.e. the excess charge $2Q$, not $3Q$ of a tri-pole or $4Q$ of a 4-in state in pyrochlores. It indeed allows one to describe more logically the motion of defects in a MF state. I have accepted this useful suggestion of the Referee and used this notion in the discussion of corresponding questions, in the modified parts of the corresponding section of the main text, and especially in the new part SI 1 of the Supplementary Material, devoted specifically to the discussion of the creation and motion of defects/excitation on MF background.

4. The point 4 raised by the Referee is indeed rather interesting and deep, especially his remark on the possible topological characteristics of such defects. This remark is actually also not an objection, but rather a question of a possible more general significance of my picture. The Referee writes: *“It is certainly interesting that scalar defects in the magnetic sector, either bookkeeping or emergent, induce vector defects in the electric sector. Is it possible to make a more systematic analysis of how these vectors are created and annihilated including what quantities are conserved and how they interact? Are they topological, do they have their own emergence associated with them independently of the magnetic charge?”* In my opinion, the basic situation here is clear. Inversion symmetry at every triangle in kagome or tetrahedron in pyrochlores is already broken in a MF state, and in kagome systems even in the usual spin-ice state: the local spin structure breaks this inversion symmetry. It is this factor which actually leads to the formation of electric dipoles. Correspondingly, certain defects also are such that they violate inversion symmetry (but some others, such as all-in states, do not; but, as argued in n.1 above, they may “release” dipoles already existing before the creation of such defects). And when they move, the magnetic charge is conserved, but the magnitude and direction of dipoles change. But the question of eventual topological characteristics of such defects is indeed a very interesting question deserving special consideration. I did not do this in this paper, but from the general point of view, or from the experience we have from the study of defects in different other ordered states, as was done e.g. in the work of Volovik and Mineev, one should expect a nontrivial topological character of defects in this case as well. I am grateful to the Referee for this question which one should indeed consider separately. At the moment I cannot say much about that, besides what I wrote above. Once again, it is an interesting question, but it lies out of scope on my present study; it should present material for special investigation.

5. In the last remark the Referee asks about the relation of dipole pairing and the existence of charge-anticharge pairs. This is again not an objection, but rather a request to discuss some points in more detail. He agrees that this result is interesting and writes *“The fact that a charge-anticharge pair in the magnetic sector generates a pair of anti-aligned electric dipoles is interesting. It would be nice to have a more detailed discussion of this absence of time reversal symmetry and where it comes from, even though this may touch on work from*

previous papers. It would also be interesting to have some insight into the role of fragmentation in this regard.” I am glad that the Referee likes this, actually the main, result of my paper. But I do not quite understand his remark about “*this absence of time reversal symmetry and where it comes from*”. Time-reversal symmetry is broken in spin ice anyway, as in any magnetic system, in all different versions of spin ice, with or without moment fragmentation. Actually I don’t see any relation of time-reversal symmetry breaking or change thereof and the appearance and eventual pairing of electric dipoles. The presence and character of these dipoles has rather much to do with the breaking of inversion symmetry, but not with the time-reversal symmetry breaking. If I would replace some magnetic ions in spin ice by nonmagnetic ones, without any magnetic moment, this could indeed modify the local time-reversal properties, and simultaneously it may influence the dipole pattern of the material; and this, in its turn, can be sensitive to the presence or absence of moment fragmentation with monopole ordering. But this is a quite special situation which might be important for some specific experiments, e.g. for nonstoichiometric or doped spin ice systems, which in fact may be quite specific for a particular situation considered. But I did not consider such situations in this paper.

As to the minor points raised in this report:

1. The Referee noticed an error in spin orientation in Fig.2(a). I myself have already noticed this error, and already corrected it. But I am grateful to the Referee for careful reading of my paper and for pointing out such technical defects.

2. As to the remark 2 of the report, indeed my expression in the text was not very accurate. What I had in mind, and I also wrote it in the text, is that the notion of moment fragmentation is nowadays used by most people, in particular practically in all experimental papers devoted to this subject, for the state with ordered monopoles. But indeed in the paper [11] it was used in a more general sense. I have now reformulated this sentence in the footnote [13], giving full credit to Ref. [11]. This sentence now reads: “As is clear from Ref. [11], such decomposition of total magnetization into divergence-free (Coulomb phase) and divergence-full part is valid even for random monopoles. However in the follow-up literature the terminology “magnetic fragmentation” is always used for the state with ordered monopoles and antimonopoles at every site (every tetrahedron in pyrochlores or every triangle in kagome systems). In this paper we will also use the term MF in this sense.”

In effect, most of the remarks of Referee 2 are now taken into account in the modifications made in the manuscript, in the main text and in the new big section SI 1 in the Supplementary Material. All modifications are marked by blue colour in the pdf file (unfortunately not well seen in the printed copy in the electric light, but clearly seen in daylight and also on the computer screen). I hope these clarifications and modifications made in the text in response to the remarks of the report would satisfy the Referee, and he would have no more objections to the publication of my paper.

Response to the report of Referee 3

I am glad that the Referee 3 “found the paper interesting, well written and the analysis convincing”. I am very grateful for the clarification of the experimental situation in a particular material $\text{Nd}_2\text{Zr}_2\text{O}_7$ and for pointing out another more appropriate example, $\text{Ho}_2\text{Ir}_2\text{O}_7$, and also for pointing out a very useful reference on the paper by E. Lhotel *et al.*, *J. Low. Temp. Phys.* (2020) which I did not know (it was published after my paper was already completed). In response to this useful information I have now somewhat changed the discussion of experimental situation in my paper, and I also added the reference to the paper pointed out, my new reference [12]. These were very useful remarks indeed, and I am very grateful to the Referee for them.

The typo mentioned is corrected.

Reviewers' Comments:

Reviewer #1:

Remarks to the Author:

I appreciate the effort the author has put into responding our questions and comments, and adapting the text accordingly. My impression after reading his answers and the new manuscript is that the results deserve to be published; however, it seems to me that its merits are not enough to do so in Nature Communications. My conclusion is based on the following.

i- I do agree with the author in his reply to Referee 2 that qualitative is not necessarily bad. As a pertinent example, the prediction of dipolar electric moments associated with magnetic monopoles (ref. [9] in the text) was a big conceptual step, it was qualitative (in the sense that no indication was given on the size of this dipolar moment), and it clearly pointed to new physical phenomena where to look for. On the other hand, the author predicts in the first part of the present work new behaviour (a reduction of electric activity when a monopole crystal with magnetic fragmentation is formed) based on the previous conceptual paper. This second step is less conceptual and has a more clear experimental motivation (which is what justifies the author to concentrate on the electric dipole pairing more than in electric currents). It thus should be more explicit in this sense; that is why I asked for more details on how to detect the studied behaviour in my previous report.

ii- Regarding this previous point, the author says: <<As the Referee mentions, there indeed may exist two situations. In one, in systems such as Dy₂Ti₂O₇ or in some kagome systems the appearance of the ordered state with moment fragmentation (MF) at low temperatures reduces the symmetry of the system and would be a real phase transition in the Landau sense. In this case the anomalies in electric response, e.g. in microwave absorption, due to the change of dipole pattern accompanying MF, should have sharp features at T_c, similar to those observed in [9].>>

I'm not aware of this first situation the author mentions, with spontaneous symmetry reduction leading to moment fragmentation in real materials. He mentions Dy₂Ti₂O₇, but in this case the only way to obtain single monopole charge order I know of is by using a magnetic field along [111]. This leads to a crossover into the Kagomé ice at low fields, and then (at very low temperatures) to a first order transition into a monopole crystal. This last case would lead to an abrupt change, easier to detect experimentally. However, this phase transition is into a fully magnetically ordered configuration (not into a moment fragmented state). Hence, it cannot be proposed as an instance of the present study. As far as I know, the best case I can mention (with spontaneous symmetry breaking into a fragmented state) has been considered only as a theoretical possibility (see Slobinsky et. al., arXiv:2011.15017), with a physical bases but with no experimental counterpart.

As I mentioned in my first report, and I think the author agreed, the second possibility would be much harder to detect. Lacking an estimation of the order of magnitude of what is expected I understand then that, at least for the known materials, it may be hard to detect the proposed electric activity reduction due to electric dipole pairing.

iii- I mentioned before that the second part of the manuscript was much vaguer than the first one; my questions (and I think some of those made by Referee 2) were directed to try to understand or redirect this. I think the answer of the author to one comment of Referee 2 explains the main spirit of this section. I quote below (the parenthesis are mine):

<<(Referee 2:) "Both unpaired dipole defects and all in or all out defects are deconfined and both have a finite energy scale for creation, so I would have expected to have a fluid of these two objects in the bulk of the sample with their densities controlled by their respective Boltzmann weights". (The author:) Of course the Referee is completely right at this point, and I have not said anything in my paper contradicting this. I just did not consider the situation with many such excitations; my aim was to investigate the electric activity, dipole character of different such

isolated defects. Of course if one would consider the situation at finite temperatures, the thermodynamic and dynamic properties of systems with finite concentration of such excitations, one should have done what the Referee writes. But, once again, my aim was quite different – it was the question which defects or excitations would have electric activity and how the dipole structure of those would look like. It is a completely different task, a different question than what the Referee mention in this remark – with which, once again, I completely agree.>>

This explains that this second part was not aimed at explaining the electric behaviour of the magnetically fragmented phase with coupled electric dipoles in thermal equilibrium, but mainly at describing the different possible defects/excitations (mainly, in isolation). Again, I believe that, given the relative importance of the conceptual contribution here and the stage of the research in the topic, the target should try have a closer link to what is observable.

In summary, I think that in its present form the manuscript makes interesting points that will be of interest to the community, but its merits are not enough to guaranty its publication in Nature Communications.

Reviewer #2:

Remarks to the Author:

The author has answered a number of questions raised by the referees and put others in context and I am satisfied with this response.

I recommend publication of this interesting article and hope the unanswered questions will generate further work.

Reviewer #3:

Remarks to the Author:

Having read the revised manuscript and supplemental material, I maintain my recommendation in favour of publication in Nature Communications. As I wrote in my previous report, I think this is a very interesting work which will inspire future developments from both experimentalists and theorists.

I think the clarifications made by the author in response to the questions raised by the other reviewers are useful. The author has not added much in the way of quantitative analysis to the manuscript, so in this respect he has arguably not addressed all of the criticisms from the other referees. Nevertheless, since I already thought the manuscript was sufficiently novel and robust even as a mostly qualitative analysis, I have maintained my recommendation.

There is one minor point which I came across which confused me when re-reading the supplemental material. The authors states that in the moment-fragmented state of the 'kagome from pyrochlore' systems neighboring triangles should have opposite circulating currents. When I tried to reproduce this result, I found that in fact neighbouring triangles have the same sense of current circulation. I have attached a pdf describing this calculation so that the author can check if I am correct, and if so revise accordingly.

In any case, it does not have serious consequences for the rest of the manuscript, since the main point of that section is that the current loops are long-range ordered in the kagome-from-pyrochlore setting, which is true either way.

Response to the second report of Referee 1

As to the longer report of Referee 1 with some critical remarks, I want to say that this report, as well as his first report, is in spirit rather positive and very constructive. His main, and actually the only objection to the paper in the second report is that in his opinion there is not enough discussion in the paper about possible experimental consequences of my results. In his opinion such discussion could strongly benefit the paper, make it more useful, and would increase its impact. I understand his point, and in principle I may agree with it (although on one point I don't agree with him, see below). In the previous version of the paper some remarks to this end were actually contained, but apparently not enough, and scattered in different parts of the text. Now, in response to his request I have expanded corresponding discussion and put it all together in one place at the end of the paper, pp. 15–17. I am actually grateful to the Referee for his insistence as to this point; it stimulated me to rethink this point and to look at it from this point of view. I hope that the respective discussion now added to the text would indeed make my paper more useful for broader audience, including, most important, experimentalists.

As to the specific remarks of the Referee, about the applicability of my treatment to $Dy_2Ti_2O_7$ and $Ho_2Ti_2O_7$, the Referee is right, these are by themselves not good examples. Actually I wrote that “*in systems such as $Dy_2Ti_2O_7$ or in some kagome systems, the appearance of the ordered state with MF at low temperatures reduces the symmetry of the system and would be a real phase transition in the Landau sense.*” I mentioned $Dy_2Ti_2O_7$ not as a real examples of a separate MF state, but rather just to contrast the situation in such systems with that where for example Ir AIAO ordering induces the formation of such state on the rare earths sublattice. But this was indeed an unlucky expression. In the new modified part with the discussion of possible experimental manifestations of my effect I have now reformulated this, removing this unlucky example, and concentrated more on kagome systems, in which the conditions for the realisation of such situation seems to be better than in pyrochlores.

One point with which I actually do not agree with Referee 1 concerns the second part of my paper, dealing with the properties of excitations, defects and domain walls (DW), including their electric activity. In a sense the Referee downplays the potential significance of this part. But in my opinion this part may be just as important as the first part – though the treatment in it is indeed based on the main effect discussed in the first part of the paper. The point is that the unique feature of a moment fragmentation (MF) state is the coexistence in one spin subsystem of an ordered and disordered components. As always, each time we have long-range ordering in a system, the most interesting effects are connected even not so much with the ground state itself – the perfectly ordered state may be even somewhat boring – but with the excitations violating this order. In my book [36] published in Cambridge in 2010 I tried to carry out this program, and it is even reflected in its title: “*Basic Aspects of the Quantum Theory of Solids: Order and Elementary Excitations*”. However just for the MF state, with its remaining spin disorder, for example the monopole DW would be rather elusive, there would be no prominent magnetic contrast on monopole domains and on the corresponding DW. Probably just because of that there was no study, even no mention of

such DW in literature. And just the fact that, as is shown in the second part of my paper, such DW (and also some point defects) would be “decorated” by electric dipoles would make their observation possible, e.g. by STM (for example using electron standing waves [37]). And the electric activity of such DW could in principle allow one to influence and control them by electric field, as was done e.g. for the ordinary DW in ferromagnets in [34]. Similarly one could think of controlling point defects with uncompensated dipoles by an inhomogeneous electric field, cf. e.g. the study of skyrmions by such method in [38]. Thus, in my opinion, just from the point of view of experimental manifestations this part of my paper can provide more possibilities to carry out corresponding experiments, some of which I now discuss in the paper.

Just one general remark. I am a theoretician, so that generally speaking I cannot know all the different experimental techniques. I think that if the main idea of the paper is clear and interesting, good experimentalists can often devise such experiments which would not come to my mind. Actually this is what happened with my previous paper on a related topic [8] in which I first suggested the idea of dipoles on monopoles. I myself suggested one experiment, the measurements of electric activity by microwave absorption, and soon after that such an experiment was indeed carried out in our institute, Ref. [12]. But for example it did not come to my mind that one can check this effect by using Raman scattering to prove the existence of such dipoles on monopoles, which was done in a cleverly-devised experiment in the paper “Experimental identification of electric dipoles induced by magnetic monopoles in $Tb_2Ti_2O_7$ ”, published in PRL in 2020 [13]. I hope that the same may be the case with the present paper. But, still, I have tried to do my best and discussed now in the new part of the text several experiments which I can envisage to check for my effect, and also discussed which materials could be the most promising for these experiments. I hope Referee 1 would find this new discussion adequate and would be satisfied with my reaction to his criticism.

Response to the second report of Referee 2

I am very glad that the second Referee, Prof. Holdsworth, now agrees with my responses to his questions/comments and with the modifications made, and that he now recommends the publication of my paper. Once again, his opinion, that of the “father” of this whole field, is very important for me, and I am glad that I managed to convince him.

Response to the second report of Referee 3

I am very grateful to Referee 3 for his/her recommendation to publish my paper, but most of all for noticing the mistake in one of the figures in the Supplementary Material. I indeed overlooked one point: I thought about spontaneous currents in individual tetrahedra (or rather in base triangles of those) which should be opposite for monopoles and antimonopoles, because they are odd functions of S . But I somehow overlooked the fact that in kagome-from-pyrochlores I considered, these tetrahedra themselves are alternating, up and down. I myself wrote in the text that the somewhat unlucky feature is that the orbital moment created by spontaneous currents is in the same direction as the total spin moment of the respective tetrahedra or triangles in pyrochlore and kagome systems. In effect the orbital moment would add only a small fraction of the total moment, so that the purely magnetic contribution of this effect would be small. That is actually why I transferred the discussion of this effect to the Supplementary Material. But when treating the kagome-from-pyrochlore I somehow overlooked this. Referee 3 is of course right, this remark is very useful, and I am very grateful to him/her for pointing this out. Of course I now corrected this drawback in Fig. S6.

Reviewers' Comments:

Reviewer #1:

Remarks to the Author:

I appreciate again the effort the author has put into responding all our questions and comments, and adapting the text accordingly.

In this new version, the author has changed a misleading statement, involving a phase transition that was not into a state with Magnetic Moment Fragmentation. Also, I think the main target of the second part of the paper, discussing different types of defects and their electric activity, is now easier to understand. Finally, the added part on experiments dealing with some of this physics (with an explicit mention of detection, and even control through some of these defects) makes the study much more interesting and less abstract.

I think the author has now answered my main concerns, raised in my first and second reports. Given the above, I will recommend this manuscript for publication in Nature Communications.